# Short-term forecasting of regional biospheric $CO_2$ fluxes in Europe using a light-use-efficiency model (VPRM, MPI-BGC version 1.2)

5 Jinxuan Chen[1], Christoph Gerbig[1], Julia Marshall[1], and Kai Uwe Totsche[2]

[1]Department Biogeochemical Systems, Max Plank Institute for Biogeochemistry, Jena, 07745, Germany
[2]Friedrich Schiller University, Jena, 07743, Germany

*Correspondence to*: Jinxuan Chen (jichen@bgc-jena.mpg.de)

10 **Abstract.** Forecasting atmospheric $CO_2$ concentrations on synoptic time scales (~days) can benefit the planning of field campaigns by better predicting the location of important gradients. One aspect of this, accurately predicting the day-to-day variation in biospheric fluxes, poses a major challenge. This study aims to investigate the feasibility of using a diagnostic light-use-efficiency model, the Vegetation Photosynthesis Respiration Model (VPRM), to forecast biospheric $CO_2$ fluxes on the time scale of a 15 few days. As input the VPRM model requires downward shortwave radiation, 2 m temperature, and Enhanced Vegetation Index (EVI) and Land Surface Water Index (LSWI), both of which are calculated from MODIS reflectance measurements. Flux forecasts were performed by extrapolating the model input into the future, i.e. using downward shortwave radiation and temperature from a numerical weather prediction (NWP) model, as well as extrapolating the MODIS indices to calculate future 20 biospheric $CO_2$ fluxes with VPRM. A hindcast for biospheric $CO_2$ fluxes in Europe in 2014 has been done and compared to eddy covariance flux measurements to assess the uncertainty from different aspects of the forecasting system. In total the range-normalized mean absolute error (normalized) of the 5 day flux forecast at daily timescales is 7.1%, while the error for the model itself is 15.9%. The largest forecast error source comes from the meteorological data, in which error from shortwave radiation 25 contributes slightly more than the error from air temperature. The error contribution from all error sources is similar at each flux observation site, and is not significantly dependent on vegetation type.

## 1 Introduction

Human activities have significantly influenced the carbon cycle of the earth system since industrialization, with the accumulation of greenhouse gases in the atmosphere leading to radiative 30 forcing and climate change (IPCC, 2014). The carbon exchange between the surface and the atmosphere still remains largely uncertain due to the complexity of processes and a lack of observations (Le Quéré et al., 2009). Therefore more measurements are needed, especially over emission hotspots and regions lacking observations. Field campaigns to measure greenhouse gases, such as research flights and measurements in remote areas, can fill the observation gap in the 35 troposphere and over regions not covered by existing networks, but they are often time-limited. To make the best use of these limited measurements, field campaigns require careful planning. An

atmospheric $CO_2$ forecast on synoptic time scales (~days) can be helpful in such cases, for it provides an estimate of what signals are expected during the experiment and a physical explanation of the observations.

The research campaign CoMet (Carbon dioxide and Methane Mission), organized by the Deutsches Zentrum für Luft- und Raumfahrt (DLR), made a series of airborne and ground-based measurements of greenhouse gases in Europe. The campaign took place from May 15th to June 12th 2018, during which four aircraft participated, including the High Altitude and LOng Range Research Aircraft (HALO) and three light aircraft. During the campaign, HALO was equipped with an Integrated Path Differential

Absorption (IPDA) Lidar (CHARM-F) (Amediek et al., 2017), and carried out nine flights with a total of 65 flight hours. Continuous online in situ $CO_2$, CO, $CH_4$ and water vapor measurements were also made onboard with the Jena Instrument for Greenhouse gas measurements (JIG) and air samples were collected with the Jena Air Sampler (JAS). The campaign performed measurements over different surfaces from northern Europe to North Africa to assess and validate the new remote sensing

instrument CHARM-F. Special attention was paid to two areas: Berlin (and nearby power plants) and the Upper Silesian basin, which are significant European point sources of $CO_2$ and $CH_4$ respectively. Ground-based and light aircraft measurements were also made in the two regions with the remote sensing instrument Methane Airborne Mapper (MAMAP) (Gerilowski et al., 2011) and portable ground-based Fourier Transform Infrared Spectrometers (FTIR) (Butz et al., 2017).

During the planning of the campaign, a $CO_2$ and $CH_4$ forecasting system was developed to support the mission; this paper focuses on the biogenic fluxes for the $CO_2$ component. The forecast provided 5 day $CO_2$ forecast fields at a fine spatial resolution (2 km x 2 km) within the observing area, and a coarser resolution over the European domain (10 km x 10 km). The forecast product is not only helpful in terms of planning observations, offering meteorology and GHG fields to capture $CO_2$/$CH_4$ plumes, but

can also provide a priori vertical information for the retrieval of remote sensing observations.
There are several existing models that can simulate atmospheric $CO_2$ on an appropriate scale, including Eulerian mesoscale models such as WRF-GHG (Beck et al., 2011;Pillai et al., 2016) and CHIMERE (Aulagnier et al., 2010). These models consist of an atmospheric tracer transport model coupled to fluxes representing the source and sink processes of $CO_2$. By providing meteorological forecast fields

and future fluxes of $CO_2$ to the model, the forecast $CO_2$ concentration fields can be obtained. The challenge of $CO_2$ forecasting comes with the provision of accurate $CO_2$ flux variations on sub-daily time scales. A global atmospheric $CO_2$ forecast system has been developed as part of the Monitoring of Atmospheric Composition and Climate – Interim Implementation (MACC-II) service (Agusti-Panareda et al., 2014;Agusti-Panareda et al., 2016). These studies have shown that although transport plays a

first order role in synoptic $CO_2$ variability, the day-to-day variability of NEE also plays an important role. Therefore it is crucial for $CO_2$ forecasts to capture the day-to-day NEE variability in real-time, instead of using climatological values.
There are many models that can simulate biospheric $CO_2$ NEE on hourly time scales (Boussetta et al., 2013; Mahadevan et al., 2008). These models can be briefly grouped into two types: process-based

models and light-use-efficiency (LUE) models. Process-based models use meteorological data as input and simulate the physiological processes of vegetation, for example BIOME-BGC (Running and Hunt

Jr, 1993), TEM (Zhuang et al., 2003) or the Carbon Exchange in the Vegetation-Soil-Atmosphere model (CEVSA) (Woodward et al., 1995). Such models usually need a number of parameters to describe the complex vegetation processes responding to meteorological drivers. The second type, LUE models, regard ecosystem gross primary production (GPP) as the product of photosynthetically active radiation (PAR), the fraction of photosynthetically active radiation absorbed by the photosynthetically active portion of the vegetation ($FAPAR_{PAV}$), and the radiation use efficiency ($\varepsilon$). Such models include the Vegetation Photosynthesis and Respiration Model (VPRM) (Xiao et al., 2004;Mahadevan et al., 2008), the MODIS Daily Photosynthesis Model (Running et al., 2000) and the Carnegie-Ames-Stanford Approach (CASA) (Potter et al., 1993).

The $CO_2$ forecast in MACC-II uses the process-based model CTESSEL to compute biospheric $CO_2$ fluxes and evapotranspiration online (Boussetta et al., 2013;Agusti-Panareda et al., 2016), which makes the two variables consistent in the forecast system. However the challenge of providing accurate $CO_2$ fluxes is due to the complexity of vegetation processes and the lack of near-real-time (NRT) observations on vegetation state. Therefore, using a LUE model for $CO_2$ flux forecasting, which is a data-driven approach having less parameters compared to process-based models, is a possible way to improve the quality of $CO_2$ fluxes in forecasting. It should be note that unlike the Copernicus Atmosphere Monitoring Service (CAMS) $CO_2$ forecasting which is operational and global, we target to build a regional $CO_2$ forecast system and only operate the forecast within a shorter period (e.g. several months). Therefore the issue of $CO_2$ budget conservation is less important comparing to a operational global forecast model.

In our case, we predict $CO_2$ fluxes based on the LUE model VPRM, which is driven by the Enhanced Vegetation Index (EVI) and the Land Surface Water Index (LSWI) as well as the meteorological variables 2 m air temperature and downward shortwave radiation. The EVI and LSWI are derived from Moderate Resolution Imaging Spectroradiometer (MODIS) reflectance data, in which the MOD09A1N product provides NRT surface reflectance data, thus the NRT observations on vegetation state can be used in flux forecasting. VPRM has a strong predictive ability for NEE while maintaining simplicity in having only four parameters for each of the seven vegetation types, which makes it suitable for our case. The flux forecast is then made by predicting the input of VPRM, for which different prediction methods were tested.

The model VPRM is one of the commonly used surface flux models in atmospheric CO2 simulations and inversions (e.g. Ahmadov et al. (2007), Pillai et al. (2016), Wu et al. (2018)). The uncertainty of the flux model is an essential question in inverse modeling Lasslop et al. (2008), and the uncertainty of 3-hourly, monthy, as well as annually integrated NEE simulated by VPRM has been well assessed by Lin et al. (2011). They established a general framework to attribute error to different sources of uncertainty (driving data, model parameter, observation and model misrepresentation). In their work the model's sensitivity to each source of uncertainty is calculated. With an estimation of errors in each variable (input data, parameter etc.), one can then attribute the total error to those uncertainty sources by multiplying the error in source with the model's sensitivity.

Back to this study, our aim is to investigate the feasibility of using such a data-driven model to predict near-future carbon fluxes. Given the uncertainties in meteorological forecasts, the near-real-time

MODIS product, and all the necessary extrapolations, it is not clear if such a model can still predict realistic carbon fluxes.

This study describes the development and assessment of a biospheric $CO_2$ flux forecast based on the LUE model VPRM, with the goal of providing accurate hourly 5-day flux forecasts. By using a hindcast and comparing the results to flux tower sites across Europe the error in the prediction is quantified, and the predictive ability of the $CO_2$ flux forecasts is assessed.

## 2 Methodology

The $CO_2$ flux forecast consists of two steps as shown in Figure 1. Model inputs are first predicted 5 days into the future, then NEE is estimated based on the standard VPRM model, using parameters optimized in previous studies (Kountouris et al., 2018). Each input which must be forecast results in corresponding errors. We systematically evaluate the flux forecasting error associated with each of these predictands.

This section describes the framework of the VPRM forecasting model for biospheric $CO_2$ fluxes, as well as the method used to evaluate the error introduced by each element of the forecast.

For the meteorological input data, we use hourly ECMWF 5 day forecasts of temperature and short wave radiation. The EVI and LSWI indices are derived from MODIS surface reflectance data. These provide the indices for an average of the past eight days, and we forecast these indices for the next five days based on linear extrapolation or persistence. We then use these predicted input data to generate NEE using VPRM.

### 2.1 VPRM data processing

### 2.1.1 Standard processing for past periods

The flux estimation is based on VPRM, a light use efficiency (LUE) model that calculates GPP with remote sensing data and meteorological data as inputs. The equation of GPP estimation is as follow:

$$GPP = \varepsilon \times FAPAR_{PAV} \times \frac{1}{1+PAR/PAR_0} \times PAR \tag{1}$$

The light use efficiency $\varepsilon$ can be decomposed as:

$$\varepsilon = \lambda \times T_{scalar} \times W_{scalar} \times P_{scalar} \tag{2}$$

Where $T_{scalar}$, $W_{scalar}$ and $P_{scalar}$ represent the temperature sensitivity of photosynthesis, the water stress effect, and the effects of leaf age on canopy photosynthesis, respectively, while $\lambda$ is an adjustable parameter in the model. $T_{scalar}$ is estimated from air temperature, and $W_{scalar}$ and $P_{scalar}$ are estimated from LSWI. See details in Mahadevan et al. 2008.

The $FAPAR_{PAV}$ in the model is estimated as a linear function of EVI, and PAR is closely correlated with downward shortwave radiation. Therefore the complete expression for GPP in VPRM is:

$$GPP = (\lambda \times T_{scalar} \times W_{scalar} \times P_{scalar}) \times EVI \times \frac{1}{1+\frac{PAR}{PAR_0}} \times PAR \tag{3}$$

While the ecosystem respiration (R) is estimated by a simple linear model:

$$R = \alpha \times T_{air} + \beta \tag{4}$$

Where $T_{air}$ is the air temperature and $\alpha$ and $\beta$ are vegetation-class-specific parameters.

The input of VPRM can be categorized into two groups: remote sensing data and meteorological data. The remote sensing data consist of EVI and LSWI at 10 km spatial resolution (the same resolution as the atmospheric transport model), where the EVI and LSWI are aggregated from MODIS surface reflectance 8-day L3 Global 500m (MOD09A1) version 6 data. It should be noted that in the forecasting model, the MODIS NRT surface reflectance data (MOD09A1N) would be used. A locally weighted least squares (LOESS) filter (alpha=0.17) is then applied to reduce the noise. The vegetation classification map that is used (SYNMAP) (Jung et al., 2006) is also a product originally derived from remote sensing. The meteorological data include air temperature at 2m and downward shortwave radiation at the surface, which are obtained from a numerical weather prediction (NWP) model product, in our case the operational forecast archive from the European Centre for Medium-Range Weather Forecasts (ECMWF). In VPRM, there are four parameters ($\lambda, PAR_0, \alpha, \beta$) for each vegetation type. Model calibration for these parameters has been done using flux measurements in Europe in 2007(Kountouris et al., 2018).

### 2.1.2 Processing for flux prediction

To use this diagnostic model in a predictive mode, we need to forecast all VPRM input variables five days into the future. Remote sensing data and meteorological data are predicted in different ways.

For the meteorological data, forecasts from a numerical weather prediction (NWP) model are needed. In this study, in order to assess the errors brought in by the meteorological forecasting, 5-day forecasts of 2-m temperature and downward shortwave radiation at the surface for each day of the year were used. The meteorological forecast is from the ECMWF operational forecast archive, with class "od" and type "fc".

As for the remote sensing data, three sources of error had to be considered: the error induced by using the NRT version of the MODIS reflectances rather than the final product, the error of estimating the value of the indices into the future, and the effect of the LOESS filter on the end value of the dataset.

We begin by describing the LOESS filter. This filter is usually applied to a full year of data, and when smoothing a truncated dataset there is an edge effect, meaning that when new data are added to the time series and the smoothing is repeated, the output at the former edge point will change slightly. In the following section we define the error caused by such an edge effect as "error due to data truncation".

Following the filtering, the smoothed data are extrapolated five days into the future, either by linear extrapolation or by assuming persistence. The optimal extrapolation method was selected after testing the error contribution of each method.

The last error source comes from the difference between MODIS NRT and the standard product. The standard product is processed with the best available ancillary, calibration, and geolocation information while changes have been made in the NRT processing to expedite the data availability (See https://earthdata.nasa.gov/earth-observation-data/near-real-time/near-real-time-versus-standard-products).

## 2.2 Uncertainty analysis

There are uncertainties in the model, in the forecast data as well as in the eddy covariance measurement, and each of these uncertainties has different impact on the final product of the flux forecast. Therefore before getting into the error quantification and model evaluation, we will briefly discuss their roles in this study.

The uncertainty in the flux measurement has to be considered before being used as the 'truth' in the model-data comparison. The uncertainty of flux measurement from eddy covariance tower and its impact on modeling has been well investigated by previous studies. Hollinger and Richardson (2005) attribute the random error in flux measurement to three reasons: The error associated with measurement system, the error associated with turbulence transport and the statistical error relating to footprint heterogeneity. They establish the method for flux measurement error estimation and analyze it on half-hourly time scale. Chevallier et al. (2012) calculate the flux measurement uncertainty on daily time scale based on hourly uncertainty estimation from (Lasslop et al., 2008), and conclude that the daily uncertainty is small comparing to the daily NEE magnitude. A similar approach is used in Broquet et al. (2013), where the uncertainty of daily flux measurement is ignored in observation-model comparison. Therefore in this study, where all comparisons are concerned with daily time scales, uncertainty from flux measurements can be neglected.

Estimating carbon fluxes with the data-driven model VPRM will result in additional uncertainties. These uncertainties are associated with uncertainties in the driving data, the misrepresentation of the LUE approach for vegetation processes, as well as the spatial representation. We treat these uncertainties as an inherent part of the model, since they will exist despite whatever 'good' data we are using to drive the model. We define these uncertainties as the VPRM 'model error', which can be quantified by comparing the flux estimation with best driving data available for VPRM to the flux measurement. This 'model error', as an inherent error in VPRM, is then chosen as a criterion for the evaluation of the forecasting result.

Lastly the error added by the flux forecasting need to be considered. As described in 2.1.2, the flux forecast is made by predicting the driving data. Such prediction has different impact on different variables, thus introducing different uncertainties. For meteorological data, they are from the Integrated Forecasting System (IFS) model of ECMWF, which will contain model error and representation error as any NWP model(Simmons et al., 1995;Simmons and Hollingsworth, 2002). Furthermore, the model error accumulates in weather forecasting, which means the further we predict into the future, the larger the error will be. As for the MODIS data, the use of NRT data and the extrapolation we apply will surely introduce uncertainties. In addition, VPRM applies LOESS filter in the MODIS data processing to reduce noise, which means the data are constrained by the neighboring information. However, when forecasting, the data can only be constrained by the past, leading to another potential error source.

Altogether, the potential error sources of this flux forecasting system are as follows: (1) the VPRM 'model error', (2) using meteorological model data rather than site-level meteorological data, (3) using ECMWF 5-day forecast meteorology, which accumulates extra error to its initial field, (4) using NRT MODIS data, (5) using LOESS filtering to smooth the MODIS data, and (6) the prediction of MODIS data. Error (6) contains two parts: (6a) EVI prediction and (6b) LSWI prediction. In the following

discussion we use the numbering (1) to (6) to denote these error sources. The 'model error' (1) defined
above is regarded as a criterion for the forecast evaluation. We define (2) to (6) as the "forecast errors",
since they are introduced by the flux forecasting. In this study, we aim to quantify the forecast error
and the error contribution from each of the error sources, then evaluate the sum of forecast errors
against the 'model error'.

In order to quantify both the model error and the forecast error, a hindcast using the $CO_2$ flux forecast
model has been done for the year 2014 for Europe. The evaluation and comparison was done at two
spatial levels: at the flux observation site level, and at the European domain level (1/8° longitude ×
1/12° latitude). The comparison at site level aims to evaluate both the model error and the forecast error
at locations with different vegetation types, while over the European domain, the aim is to investigate
the spatial pattern of each forecast error term.

The surface $CO_2$ flux observation data comes from eddy covariance tower measurements from the
FLUXNET2015 tier one (open data) dataset (Baldocchi et al., 2001). Thirty-three European
observation sites for which both MODIS data and flux measurements for 2014 are available were
selected for data-model comparison. The selected sites' ID, location, vegetation type, and their data
DOI are listed in table 1.

To test the error contribution of the model and the 5 day flux forecast, experiments using the VPRM
forecast model were carried out to evaluate the error contribution from different sources separately, as
shown in Table 2. A control simulation and six experimental simulations (simulations a to f) were
conducted. Although the $CO_2$ flux forecast targets hourly flux prediction for the next 5 days, model
error and forecast error were analyzed on a daily time scale, as this scale is more relevant for synoptic
$CO_2$ variability in the atmosphere.

The control simulation uses standard VPRM as a reference model with "perfect" input, meaning the
MODIS EVI and LSWI standard products as well as shortwave radiation and temperature observed at
the flux site. By comparing the modeled NEE to flux measurements, we can estimate the VPRM model
error (1).

The experimental simulations a to f then included the error sources (2) to (6) in the VPRM model input
data separately, and these are compared to the reference simulation in order to isolate the individual
error contributions. The experiments aim to estimate the upper limit of forecast error, therefore in
simulations b and f, 96 h to 120 h meteorological forecasts, i.e. the last day (5th) of a 5-day forecast,
were used for each day of the year. For simulations d and f, since the MODIS EVI and LSWI products
has an 8-day period, MODIS data were first linearly interpolated to a daily scale. Then for each day of
the year, MODIS data on the $n^{th}$ day were predicted from data on the $n-5^{th}$ day.

There is a challenge in simulation e in that there are no archived NRT data for 2014, thus it is
impossible to have a comparison on the same basis with the other simulations. Instead we look at the
model's sensitivity of NEE to EVI and LSWI bias, and also compare the NRT EVI and LSWI, which
we archived from February to June in 2018 for 120 days, to the standard MODIS product over the same
period. In this way we were able to estimate the magnitude of the NRT indices' error and its impact on
the model's output NEE.

In order to make the 33 different site results comparable, the simulation output NEE was first

aggregated to daily averages, and then normalized by the range (i.e. the difference between maximum and minimum) of annual NEE at each site. The $Bias_{NEE}$, which is defined as the output NEE from the experimental simulation minus the same variable from the reference model, was then calculated and normalized by the same scalar at each site. By applying such a normalization, positive and negative NEE keep their sign, and the normalized $Bias_{NEE}$ represents a fractional bias compared to the range of annual variation. (For example a normalized $Bias_{NEE}$ of 0.1 means that the magnitude of the bias equals 10% of the annual variation.) Similar to $Bias_{NEE}$, $Bias_{-GPP}$ and $Bias_R$ are also calculated as a measure for error in simulated GPP and R. $Bias_{-GPP}$ (or $Bias_R$) is the difference of -GPP (or R) in experimental and reference simulation, normalized by the annual range of observed NEE at each site (note that the sign of GPP is reversed). $Bias_{-GPP}$ and $Bias_R$ use the same normalization scalar so that they are additive and comparable to $Bias_{NEE}$, Based on these definitions, we have:

$$Bias_{NEE} = Bias_{-GPP} + Bias_R \tag{5}$$

Thus the metrics $Bias_{-GPP}$ and $Bias_R$ represent the fractional bias of photosynthetic and non-photosynthetic part in NEE. The mean of the absolute $Bias_{NEE}$ will be the mean absolute error (MAE), which is also used as a measure for error in this research. An example of such normalization is shown for the station BE-Bra in Figure 2.

## 3 Results and Discussion

### 3.1 Error attribution on site level

By comparing the NEE output from each experimental simulation, the impact of each error source on flux forecasting can be isolated and evaluated. The normalized mean absolute error (MAE) of NEE at all 33 sites is presented in Table 3. The MAE of the total forecast error is 0.071, which is smaller than the VPRM model error of 0.159. This indicates that the forecast model is reasonably capable of predicting fluxes on diurnal time scales.

### 3.1.1 Meteorological error

Among all forecast errors, the meteorological error accounts for the largest contribution. The meteorological error can be decomposed into (2) analysis error and (3) meteorological forecast error. The former corresponds to using meteorological analysis rather than observational data, while the latter comes from the numerical meteorological forecasting, and can be estimated by comparing simulations b and a. The analysis error and meteorological forecast error are of the same order of magnitude, namely 0.046 and 0.065 respectively.

The meteorological error is then analyzed further by dividing it into the photosynthetic part ($Bias_{-GPP}$) and the non-photosynthetic respiration part ($Bias_R$) as described in section 2.2. The bias distributions of 33×365 data points are shown in Figure 3.

In figure 3, panels (a), (b) and (c) share the same x-axis, and the bias in the y-axes can be combined as $Bias_{NEE}= Bias_{-GPP} + Bias_R.$ Because a positive GPP bias will lead to a negative NEE bias, -GPP is used here to show its contribution to NEE. $Bias_{-GPP}$ has a larger vertical spread towards negative values, which means a systematic bias in GPP. In contrast $Bias_R$ is basically symmetric about zero, which

implies that the errors in temperature are random.

This seems to suggest that $Bias_{NEE}$ has a larger contribution from the photosynthetic part than the non-photosynthetic part. Knowing that $Bias_{NEE}$ is the result of biases in the two meteorological variables used in the simulation, air temperature and downward shortwave radiation (SW), we conduct two further experiments, b.1 and b.2, to quantify the error contribution from these variables separately. In b.1 only the shortwave radiation is taken from the 5-day forecast, while all other variables are from the control simulation. b.2 is similar to b.1, but in this case the forecast value is used only for the 2-m air temperature. Figure 4 shows the bias distribution of the two experiments, in which the vertical spread of bias in b.1 (a) is slightly larger than b.2 (b). The overall normalized MAE compared to the control simulation, is 0.053 when using forecast SW (b.1), while it is 0.042 when using forecast 2-m air temperature (b.2). Thus the error contribution resulting from forecast errors in downward shortwave radiation at the surface is found to be slightly larger than the error from 2-m air temperature.

### 3.1.2 MODIS error

The MODIS error consists of three parts: using NRT products, using extrapolated indices, and using truncated time series. These are tested in simulations c, d and e respectively. In general, the MODIS error is less important than the meteorological error, and the errors due to data truncation, EVI extrapolation and LSWI extrapolation result in errors of similar magnitude: 0.015, 0.013 and 0.010 respectively.

As described in section 2.1.1, the MODIS input data first need to be smoothed by a LOESS filter to reduce the noise. LOESS performs a local regression on the time series. Because the point at the end of the time series lacks a constraint from future data, it results in an error when the data are truncated. This error source is evaluated in simulation c, where for each 8 day value, only data before this time are filtered. Thus the only difference between simulation c and the reference simulation is whether each MODIS-derived index is constrained by all local data or only constrained by preceding data. Comparing simulation c and the reference simulation finds that the error due to lack of constraint from future MODIS data introduces a MAE of 0.015.

For MODIS data extrapolation, different methods were tested in an attempt to minimize forecast error. Climatological values of EVI and LSWI were considered, but they lack the advantage of a data-driven approach for realistic estimation. After testing various alternatives, two simple methods were considered: linear extrapolation based on the last three data points and persistence (assuming the indices stay the same for the next five days). Figure 5 shows the NEE bias distribution by using linear extrapolation or persistence to predict EVI and LSWI. For both indices, using the assumption of persistence results in a smaller error. The biases for the two extrapolation methods have similar distributions, but there are more outliers for linear extrapolation. This is due to the fact that linear extrapolation results in larger errors when the data are fluctuating.

Finally, the difference between using MODIS NRT data and standard data has to be considered. This includes the effect of using different attitude and ephemeris data in processing, as well as using different ancillary data products for the Level 2 processing. For L2 Land Surface Reflectance data, National Oceanic and Atmospheric Administration Global Forecast System (GFS) ancillary product are

used instead of the Global Data Assimilation System (GDAS) products used in the standard processing
      (This is described at NASA's Land, Atmosphere Near real-time Capability for EOS (LANCE) website
      https://earthdata.nasa.gov/earth-observation-data/near-real-time/near-real-time-versus-standard-
      products).

      This presented a challenge, as no MODIS NRT data were archived for the test year 2014. Thus it was
impossible to carry out a similar error evaluation as was done for other error sources. Therefore we first
      use NRT EVI and LSWI that we archived for 120 days from February to June 2018 to calculate the
      MAE of the two indices to standard products at all flux sites. The MAE of NRT EVI and LSWI for all
      sites are 0.018 and 0.026 respectively. Considering the mean EVI and LSWI, which are 0.21 and 0.11
      during this period, the magnitude of NRT EVI error is less than 10% of EVI's magnitude while the
number is 24% for the magnitude of NRT LSWI error.

      The impact of the errors in these NRT indices on the model is determined by the model's sensitivity to
      EVI and LSWI. To investigate this sensitivity, we use the result from simulation d and the reference
      simulation, and look at the difference in input EVI and LSWI, and the corresponding difference in
      output NEE. The model's sensitivity is different during the growing and the non-growing seasons, as in
the non-growing season there would be no vegetation production anyway from a slight change of EVI
      and LSWI.

      Therefore the model sensitivity is analyzed for each season separately, as shown in table 4. Difference
      in indices and the corresponding difference in daily NEE are applied with linear regression, and the rate
      of the linear function is regarded as model sensitivity. The maximum sensitivity for both EVI and
LSWI is in summer, with -9.11 [$\mu$mole m$^{-2}$ s$^{-1}$ EVI$^{-1}$] and -6.29 [$\mu$mole m$^{-2}$ s$^{-1}$ LSWI$^{-1}$] respectively.
      By assuming that the 120 days of archived NRT data is representative for MODIS NRT error, we can
      estimate the upper limit of forecasting error (4), as it is shown in Figure 6. The normalized NEE error
      in figure 6 is calculated by using MODIS NRT error times the model sensitivity, and then normalized
      by the same scalar used in previous analysis at each site. Therefore the error here is comparable to the
MAE in table 3 if we assume the MODIS NRT data in the year 2014 and 2018 have similar error
      structure. The NEE error for all sites due to NRT-EVI and NRT-LSWI are 0.024 and 0.025
      respectively, which is still smallcompared to the meteorology error in table 3.

### 3.1.3 VPRM model error

      Unlike the forecast error discussed above, the $Bias_{NEE}$ of (1) model error (reference model minus
observation) distribution of the VPRM model error is asymmetric, as shown in Figure 7. The model
      bias shows a negative correlation, which means a weaker uptake during the growing season and a
      weaker respiration during the non-growing season. Data with negative normalized NEE also
      correspond to a larger bias, which refers to larger model uncertainty during the growing season. The
      MAE of the model error is 0.166.

### 3.1.4 Errors at each flux observation site

      The MAE is also calculated at each flux measurement site and clustered according to vegetation types,
      shown in figure 8. Generally the VPRM model error (grey) is larger or similar to the forecast error

(blue), consistent with Table 3. Moreover the forecast error does not differ significantly over different vegetation types. Figure 9 shows the error contribution from each source, the meteorological error (error (2) in dark blue and error (3) in light blue) is the dominant contributor at each site, and has a similar contribution for different vegetation types. The data truncation error (4) has a stronger influence on some grass sites, because EVI at these sites is highly variable, possibly due to mowing and re-growing during the growing season. Overall, except for the data truncation error, all forecast error sources have a similar impact on each flux observation site. This shows that the forecast ability does not vary over different vegetation types.

**3.2 Spatial pattern of forecast error**

The forecast errors are also tested on the European domain from March to June (the season over which the CoMet campaign took place) in 2014, to analyze its spatial patterns. Three experiments have been done to represent the meteorological error (including analysis error and meteorological forecast error), the MODIS error (including extrapolation error and data truncation error) and the total forecast error (a combination of meteorological error and MODIS error). Figure 10 shows the mean VPRM NEE during the period and the corresponding spatial distribution of each error (in MAE).

By comparing Figures 10(a) and 10(b), it can be seen that the MAE of the total forecast error has a strong spatial relationship with the mean NEE, which indicates that the forecast error has a similar impact in all places. On a spatial level, the meteorological component still dominates compared to the MODIS error.

In the context of atmospheric $CO_2$ forecasting, the forecast $CO_2$ concentrations that are influenced by fluxes from larger MAE areas (northern France, Germany and the Balkans) may have a larger bias due to poorer flux prediction in these areas.

The flux budget over the European domain was also calculated and is shown in Figure 11. The carbon budget of the flux forecast model (in dark blue) is close to the original VPRM model (in grey), thus we are able to confidently use this flux forecast model in the atmospheric GHG concentration forecasting system and predict reasonable $CO_2$ concentrations on synoptic time scales.

As mentioned in the introduction, we are aiming for not only a flux forecast, but finally an atmospheric GHG concentration forecasting system. While this study has quantified how each error source affects the predicted biospheric fluxes, the next step is to use such predicted fluxes in an atmospheric transport model run in forecast mode, and to assess the prediction error from each source in concentration space.

**4 Conclusions**

Based on the VPRM model, we developed a forecasting model that can predict biospheric NEE for the next five days, and assess the error contribution from each aspect of forecasting. This $CO_2$ flux forecast model is a crucial component in an atmospheric $CO_2$ forecasting system, in which hourly to day-to-day $CO_2$ flux variability plays an important role. The forecast model inputs are MODIS near-real-time EVI and LSWI, as well as shortwave radiation and temperature from a meteorological forecast model. The error attribution shows that the dominant error is related to the meteorological data. We further

attribute this error to the uncertainties in forecast shortwave radiation and temperature, and found that the forecast shortwave radiation contributes slightly more to the meteorological error. Error from MODIS inputs is less important, and using a persistence assumption to predict MODIS indices resulted in smaller errors than a linear extrapolation. Overall the forecasting system error has a MAE of 0.071, which makes the model capable of forecasting $CO_2$ fluxes on the target time scale. The error

contribution is insensitive to vegetation type and consistent over the whole EU domain. The error of the forecasting system is less than the VPRM model error at flux observation site level, which means that the system performs sufficiently well for its predictive task. From the spatial distribution of the error, the absolute flux errors are larger in northern France, Germany and the Balkans, which will lead to larger bias in atmospheric $CO_2$ forecasting system. The assessment of these (and other) errors in

concentration space, using measurements from the CoMet mission as reference data, is foreseen as a follow-up study.

*Code and data availability*. The code for forecast VPRM model and the model outputs are available from http://dx.doi.org/10.17617/3.2d. The code used for model assessment and figure plotting in this

paper is also included in the same repository. The flux measurement data can be acquired from FLUXNET2015 database (see DOI in table 1). The MODIS reflectance data can be acquired from NASA's Earth Science Data Systems (https://earthdata.nasa.gov/). The ECMWF meteorology data can be retrieved using ECMWF's Meteorological Archival and Retrieval System (MARS, https://confluence.ecmwf.int/display/UDOC/MARS+user+documentation).


*Author contribution.* The experiments were planned by C. Gerbig, J. Marshall, K.U. Totsche and J. Chen. C. Gerbig prepared the standard VPRM model. J. Chen made the forecast model and performed the model simulation and assessment. J. Marshall extensively commented and revised the manuscript. J. Chen prepared the manuscript with contribution from all co-authors.


*Competing interests*. The authors declare no competing interests. *Acknowledgements.* We acknowledge funding for the CoMet campaign by MPG (Max Planck society) and by BMBF (German Federal Ministry of Education and Research) through AIRSPACE (FK 01LK1701C), and the PhD project funding from the International Max Planck Research School for Global Biogeochemical Cycles

(IMPRS-gBGC). We acknowledge the use of data products from the Land, Atmosphere Near real-time Capability for EOS (LANCE) system operated by NASA's Earth Science Data and Information System (ESDIS) with funding provided by NASA Headquarters. We acknowledge ECMWF for providing access to the ECMWF's archived data. This work used eddy covariance data acquired and shared by the FLUXNET community, including these networks: AmeriFlux, AfriFlux, AsiaFlux, CarboAfrica,

CarboEuropeIP, CarboItaly, CarboMont, ChinaFlux, Fluxnet-Canada, GreenGrass, ICOS, KoFlux, LBA, NECC, OzFlux-TERN, TCOS-Siberia, and USCCC. The ERA-Interim reanalysis data are provided by ECMWF and processed by LSCE. The FLUXNET eddy covariance data processing and harmonization was carried out by the European Fluxes Database Cluster, AmeriFlux Management

Project, and Fluxdata project of FLUXNET, with the support of CDIAC and ICOS Ecosystem
Thematic Center, and the OzFlux, ChinaFlux and AsiaFlux offices.

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

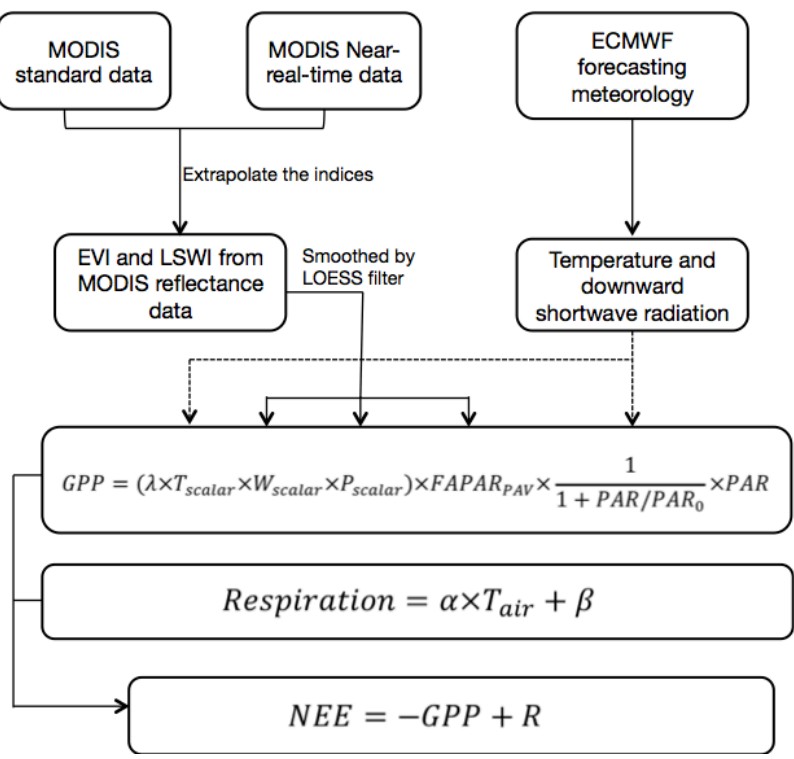

**Figure 1: Diagram of the VPRM forecasting system. The top two levels show the drivers which are predicted into the future, while the bottom three boxes are based on the standard VPRM model (Mahadevan et al., 2008).**






**Table 1: The selected FLUXNET2015 sites used for data-model comparison in this research.**

| Site ID | Latitude | Longitude | Vegetation types in VPRM | Data DOI | Reference |
|---------|----------|-----------|--------------------------|----------|-----------|
| BE-Bra | 51.3092 | 4.5206 | Mixfrst | 10.18140/FLX/1440128 | (Janssens et al.) |
| BE-Lon | 50.5516 | 4.7461 | Crop | 10.18140/FLX/1440129 | (Moureaux et al., 2006) |
| BE-Vie | 50.3051 | 5.9981 | Mixfrst | 10.18140/FLX/1440130 | (Aubinet et al., 2001) |
| CH-Cha | 47.2102 | 8.4104 | Grass | 10.18140/FLX/1440131 | (Merbold et al., 2014) |
| CH-Dav | 46.8153 | 9.8559 | Evergreen | 10.18140/FLX/1440132 | (Zielis et al., 2014) |
| CH-Fru | 47.1158 | 8.5378 | Grass | 10.18140/FLX/1440133 | (Imer et al., 2013) |
| CH-Lae | 47.4781 | 8.365 | Mixfrst | 10.18140/FLX/1440134 | (Etzold et al., 2011) |
| CH-Oe2 | 47.2863 | 7.7343 | Crop | 10.18140/FLX/1440136 | (Dietiker et al., 2010) |
| CZ-wet | 49.0247 | 14.7704 | Grass | 10.18140/FLX/1440145 | (Dušek et al., 2012) |
| DE-Akm | 53.8662 | 13.6834 | Grass | 10.18140/FLX/1440213 | (Bernhofer et al.) |
| DE-Geb | 51.1001 | 10.9143 | Crop | 10.18140/FLX/1440146 | (Anthoni et al., 2004) |
| DE-Gri | 50.9495 | 13.5125 | Grass | 10.18140/FLX/1440147 | (Prescher et al., 2010) |
| DE-Kli | 50.8929 | 13.5225 | Crop | 10.18140/FLX/1440149 | (Prescher et al., 2010) |
| DE-Obe | 50.7836 | 13.7196 | Evergreen | 10.18140/FLX/1440151 | (Bernhofer et al.) |
| DE-RuR | 50.6219 | 6.3041 | Grass | 10.18140/FLX/1440215 | (Post et al., 2015) |
| DE-RuS | 50.8659 | 6.4472 | Crop | 10.18140/FLX/1440216 | (Mauder et al., 2013) |
| DE-SfN | 47.8064 | 11.3275 | Grass | 10.18140/FLX/1440219 | (Hommeltenberg et al., 2014) |
| DE-Spw | 51.8923 | 14.0337 | Grass | 10.18140/FLX/1440220 | (Bernhofer et al.) |
| DE-Tha | 50.9636 | 13.5669 | Evergreen | 10.18140/FLX/1440152 | (GrüNwald and Bernhofer, 2007) |
| DK-Sor | 55.4859 | 11.6446 | Decid | 10.18140/FLX/1440155 | (Pilegaard et al., 2011) |
| FI-Hyy | 61.8475 | 24.295 | Evergreen | 10.18140/FLX/1440158 | (Suni et al., 2003) |
| FI-Sod | 67.3619 | 26.6378 | Evergreen | 10.18140/FLX/1440160 | (Thum et al., 2007) |
| FR-Fon | 48.4764 | 2.7801 | Decid | 10.18140/FLX/1440161 | (Delpierre et al., 2016) |
| FR-Pue | 43.7414 | 3.5958 | Evergreen | 10.18140/FLX/1440164 | (Rambal et al., 2004) |
| IT-BCi | 40.5238 | 14.9574 | Crop | 10.18140/FLX/1440166 | (Vitale et al., 2016) |
| IT-CA1 | 42.3804 | 12.0266 | Decid | 10.18140/FLX/1440230 | (Sabbatini et al., 2016) |
| IT-CA2 | 42.3772 | 12.026 | Crop | 10.18140/FLX/1440231 | (Sabbatini et al., 2016) |
| IT-CA3 | 42.38 | 12.0222 | Decid | 10.18140/FLX/1440232 | (Sabbatini et al., 2016) |
| IT-Col | 41.8494 | 13.5881 | Decid | 10.18140/FLX/1440167 | (Valentini et al., 1996) |
| IT-Cp2 | 41.7043 | 12.3573 | Evergreen | 10.18140/FLX/1440233 | (Fares et al., 2014) |
| IT-Isp | 45.8126 | 8.6336 | Decid | 10.18140/FLX/1440234 | (Ferréa et al., 2012) |
| IT-Lav | 45.9562 | 11.2813 | Evergreen | 10.18140/FLX/1440169 | (Marcolla et al., 2003) |
| IT-Tor | 45.8444 | 7.5781 | Grass | 10.18140/FLX/1440237 | (Galvagno et al., 2013) |


|  | MODIS indices | Meteorology data | Error sources |
|---|---|---|---|
| Reference simulation | Standard MODIS products | Flux site observation | (1) |
| Simulation a | Standard MODIS products | ECMWF 12h forecasting | (1)+(2) |
| Simulation b | Standard MODIS products | ECMWF 5th day forecasting | (1)+(2)+(3) |
| Simulation c | Truncated MODIS indices | Flux site observation | (1)+(5) |
| Simulation d | MODIS prediction based on fully filtered data | Flux site observation | (1)+(6) |
| Simulation e | NRT MODIS indices | Flux site observation | (1)+(4) |
| Simulation f | MODIS prediction based on truncated data | ECMWF 5th day forecasting | (1)+(2)+(3)+(5)+(6) |

**Table 2: The experiment setup and the error sources addressed in each simulation. The numbering in the last column corresponds to the error from (1) the VPRM model, (2) the meteorological analysis, (3) the meteorological forecast, (4) the MODIS NRT data, (5) data truncation and (6) the prediction of MODIS indices.**


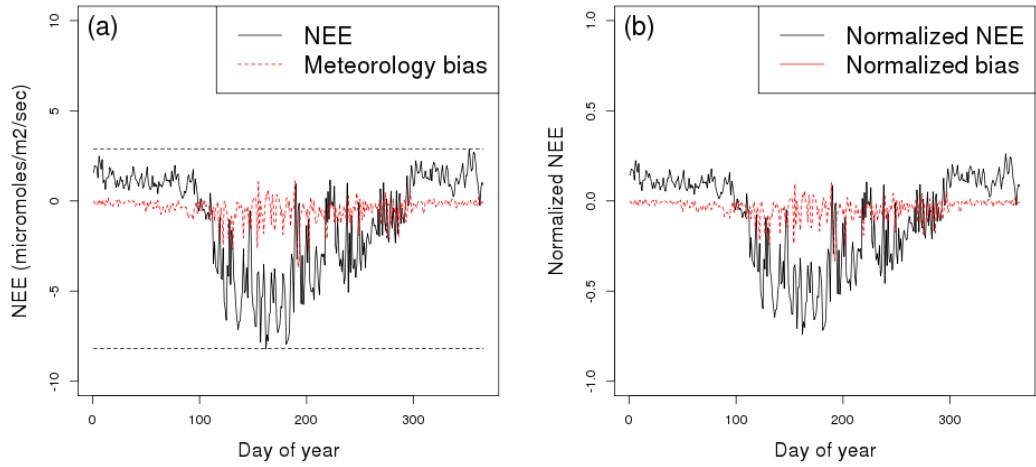


**Figure 2: Example of the data normalization at station BE-Bra: (a) NEE output from simulation a, and the corresponding $Bias_{NEE}$. The dashed black lines show the range of annual NEE. (b) NEE and bias after normalization by the range, conserving the physical meaning (release and uptake) of the sign.**



| Normalized Mean Absolute Error (MAE) for each error source | | |
|---|---|---|
| **Compared objects** | **Error sources** | **MAE** |
| a-ref. | (2) Meteorological analysis | 0.046 |
| b-a | (3) Meteorological forecast | 0.040 |
| b-ref. | (2)+(3) Meteorological error | 0.065 |
| c-ref. | (5) Data truncation | 0.015 |
| d-ref. | (6a-i) Linear EVI | 0.016 |
| d-ref. | (6a-ii) Persistence EVI | 0.013 |
| d-ref. | (6b-i) Linear LSWI | 0.012 |
| d-ref. | (6b-ii) Persistence LSWI | 0.010 |
| f-ref. | (2)+(3)+(5)+(6a-ii)+(6b-ii) Forecast error | 0.071 |
| ref.-obs. | (1) Model error | 0.159 |

**Table 3: Normalized Mean Absolute error (MAE) of NEE for each error source. The compared objects are simulation a to f, the reference simulation (ref.) and FLUXNET observation (obs.). Error sources (1) to (6) described in 2.2 can be isolated by calculating the MAE between different simulations.**


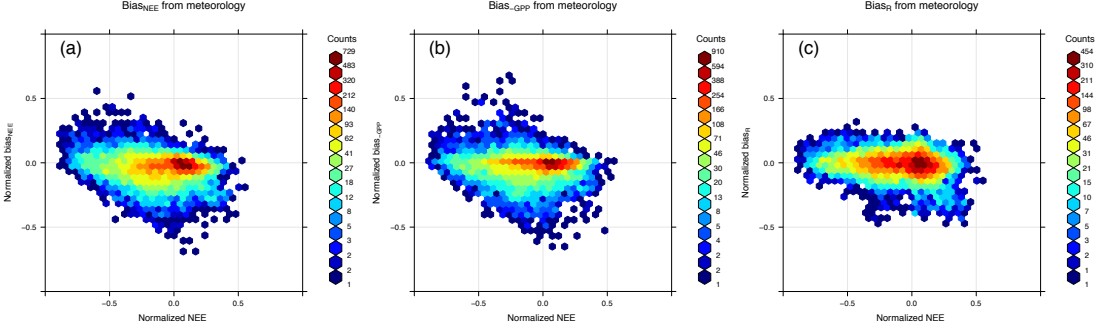

**Figure 3: (a) Distribution of normalized $Bias_{NEE}$ due to meteorological error. The x-axis refers to the normalized NEE, and the y-axis refers to the corresponding $Bias_{NEE}$ defined in section 2.2. Panels (b) and (c)**
**share the same x-axis with (a), but have $Bias_{-GPP}$ and $Bias_R$ in y-axis instead. The three biases combine as $Bias_{NEE} = Bias_{-GPP} + Bias_R$, suggesting larger contribution from photosynthetic part $bias_{-GPP}$, which is controlled by the radiation parameter rather than temperature.**

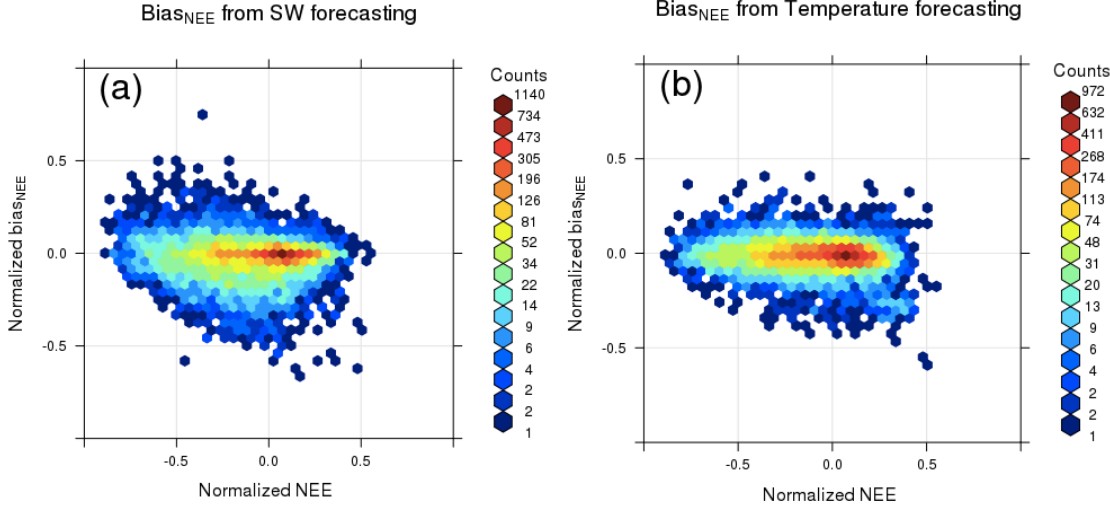

**Figure 4: The _Bias_$_{NEE}$ distribution of experiment b.1 (left) and b.2 (right). In experiment b.1 only SW is from 5-day forecast while other variables are the same with the reference simulation; while in experiment b.2 it is air temperature that only comes from 5-day forecast. The MAEs to the reference experiment are 0.053 and 0.042 respectively.**



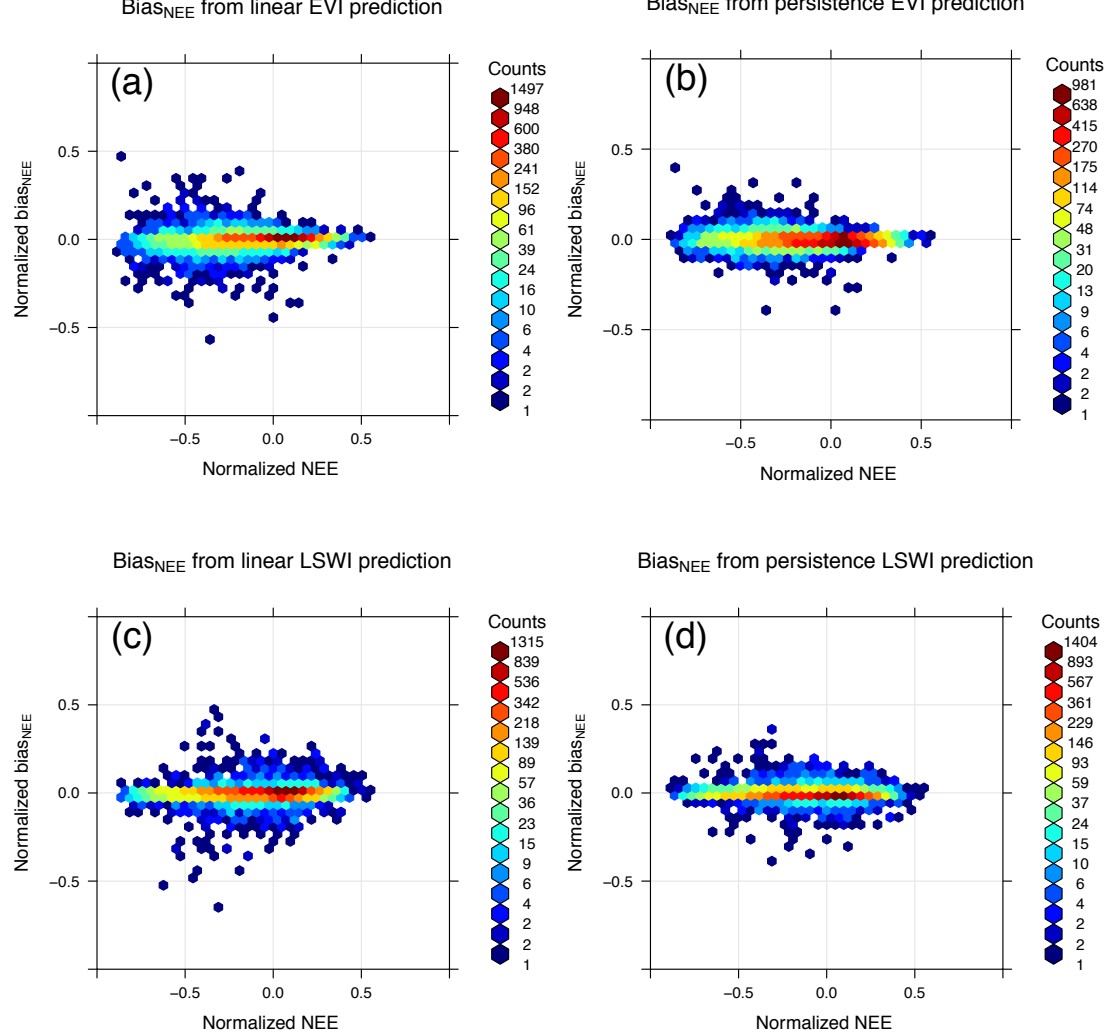

**Figure 5:** *Bias$_{NEE}$* **distribution of using linear extrapolation or persistence to predict EVI and LSWI. The persistence prediction introduces less bias than linear extrapolation for both EVI and LSWI. Therefore persistence is used in the final forecast.**


| | NEE sensitivity to EVI | | | NEE sensitivity to LSWI | |
|---|---|---|---|---|---|
| Seasons | Sensitivity [μmole m$^{-2}$ s$^{-1}$ EVI$^{-1}$] | R$^2$ | Seasons | Sensitivity [μmole m$^{-2}$ s$^{-1}$ LSWI$^{-1}$] | R$^2$ |
| Dec - Feb | -0.90 | 0.27 | Dec - Feb | -0.57 | 0.28 |
| Mar - May | -7.96 | 0.64 | Mar - May | -3.41 | 0.51 |
| Jun - Aug | -9.11 | 0.74 | Jun - Aug | -6.29 | 0.58 |
| Sep - Jan | -2.70 | 0.35 | Sep - Jan | -1.16 | 0.29 |

**Table 4: The model's sensitivity of NEE to EVI/LSWI for four seasons. The result of simulation d is used in the sensitivity calculation. Linear regression is applied to the change in EVI and the change in corresponding NEE, the maximum sensitivity appears in summer, with a slope of -10.73 [μmole m-2 s-1 EVI-1] for EVI and -6.29 [μmole m-2 s-1 LSWI-1] for LSWI respectively.**

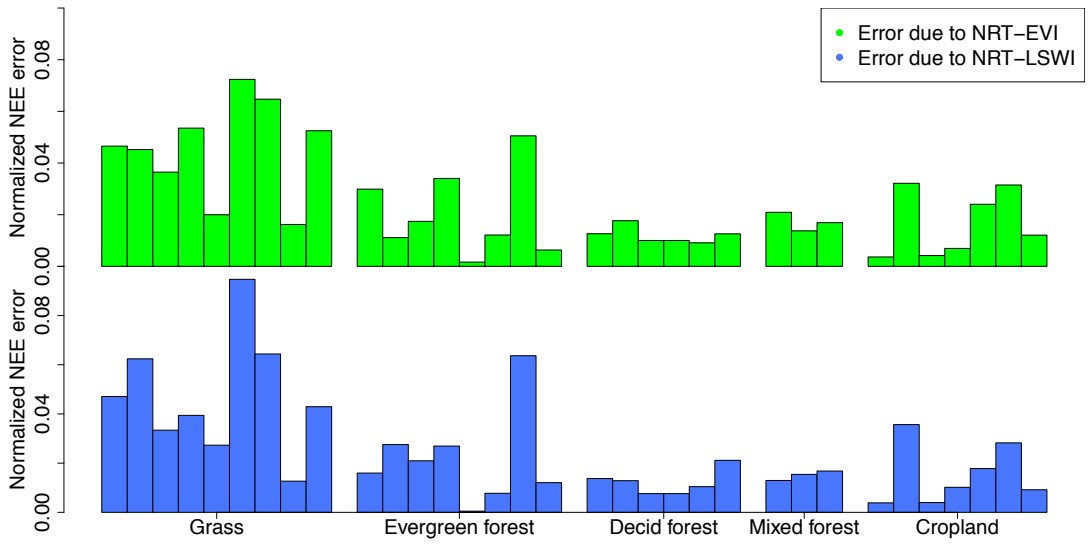


**Figure 6: The normalized error of NEE as a result of MODIS NRT error at 33 sites. 120 days from February to June in the year 2018 of MODIS NRT data are used to first calculate the EVI/LSWI differences, then times the sensitivities in table 4 and normalized by the same scalar in the previous research. The flux sites in x-axis are sorted by vegetation type and FLUXNET site-ID (from left to right:**

**CH-Cha, CH-Fru, CZ-wet, DE-Akm, DE-Gri, DE-RuR, DE-SfN, DE-Spw, IT-Tor, CH-Dav, DE-Obe, DE-Tha, FI-Hyy, FI-Sod, FR-Pue, IT-Cp2, IT-Lav, DK-Sor, FR-Fon, IT-CA1, IT-CA3, IT-Col, IT-Isp, BE-Bra, BE-Vie, CH-Lae, BE-Lon, CH-Oe2, DE-Geb, DE-Kli, DE-Rus, IT-BCi, IT-CA2).**

Bias<sub>NEE</sub> from (1)model error

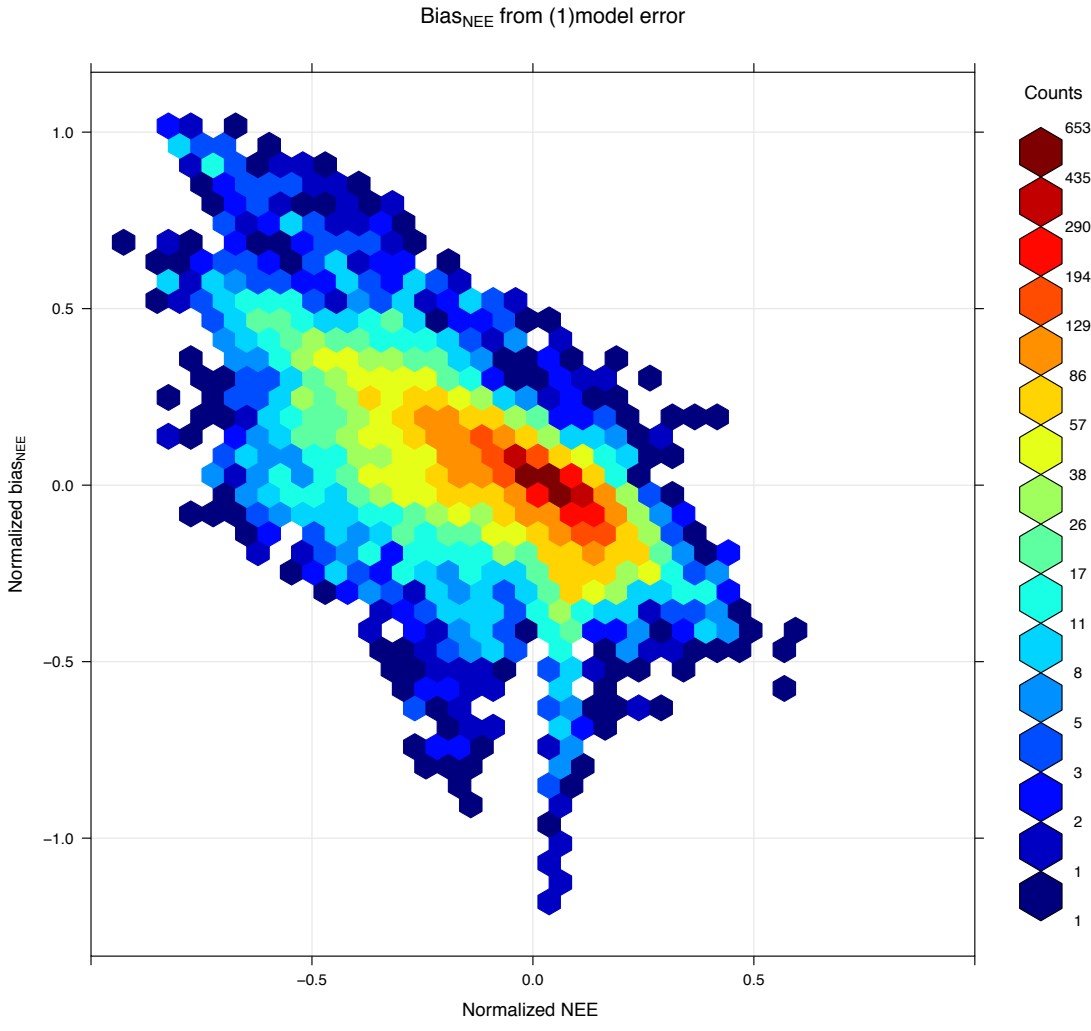

**Figure 7: The *Bias<sub>NEE</sub>* distribution of the VPRM model error.**


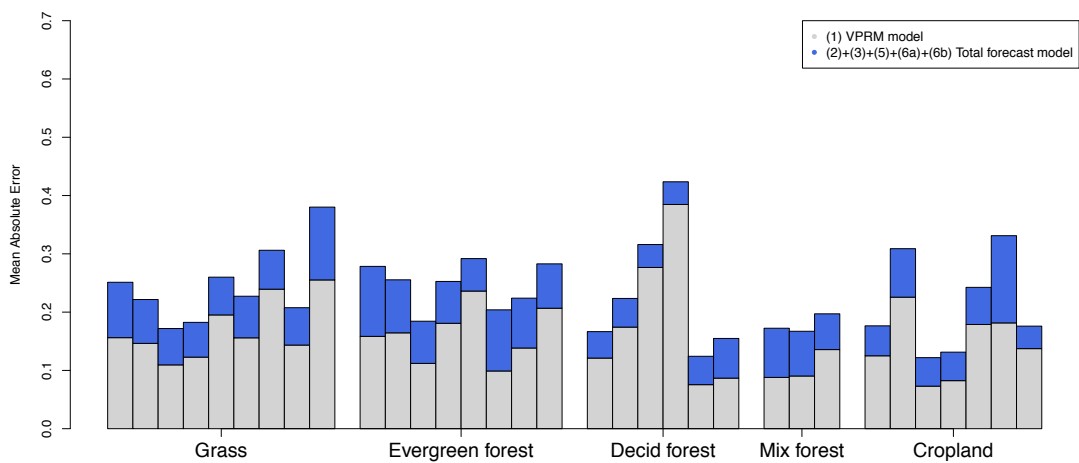

**Figure 8: Mean absolute error of the forecast error compared to the VPRM model error at each flux observation site. The model error (1) is generally larger than the total forecast error (2) to (6), and the forecast error does not differ significantly across vegetation types. The order of the flux site is the same as in figure 6.**

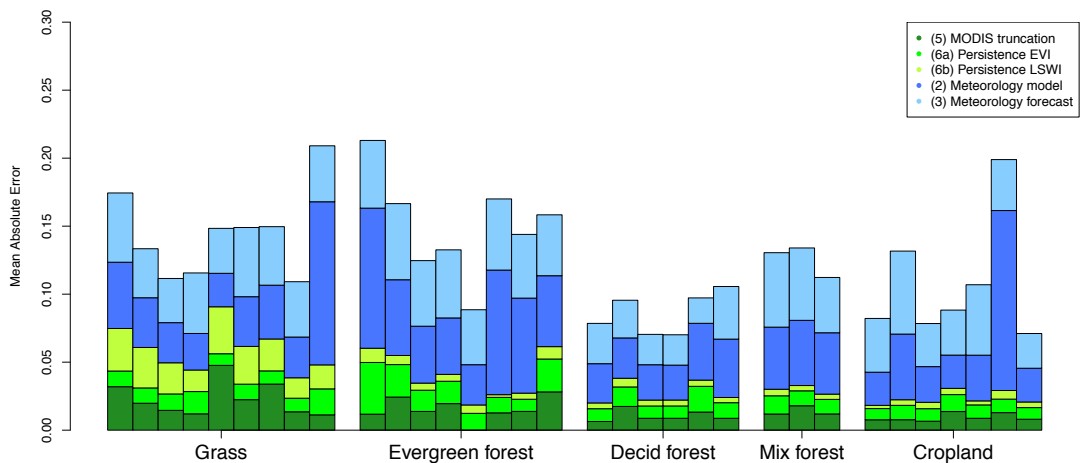

**Figure 9: Mean absolute error for different error sources at each flux observation site. The meteorological error ((2)meteorological model + (3)meteorological forecast) is the dominant contributor at each site, and has a similar contribution for different vegetation types. The data truncation error (4) has a stronger influence on some grass sites, likely due to the highly EVI variability resulting from mowing and regrowth during the growing season. The order of the flux site is the same as in figure 6.**

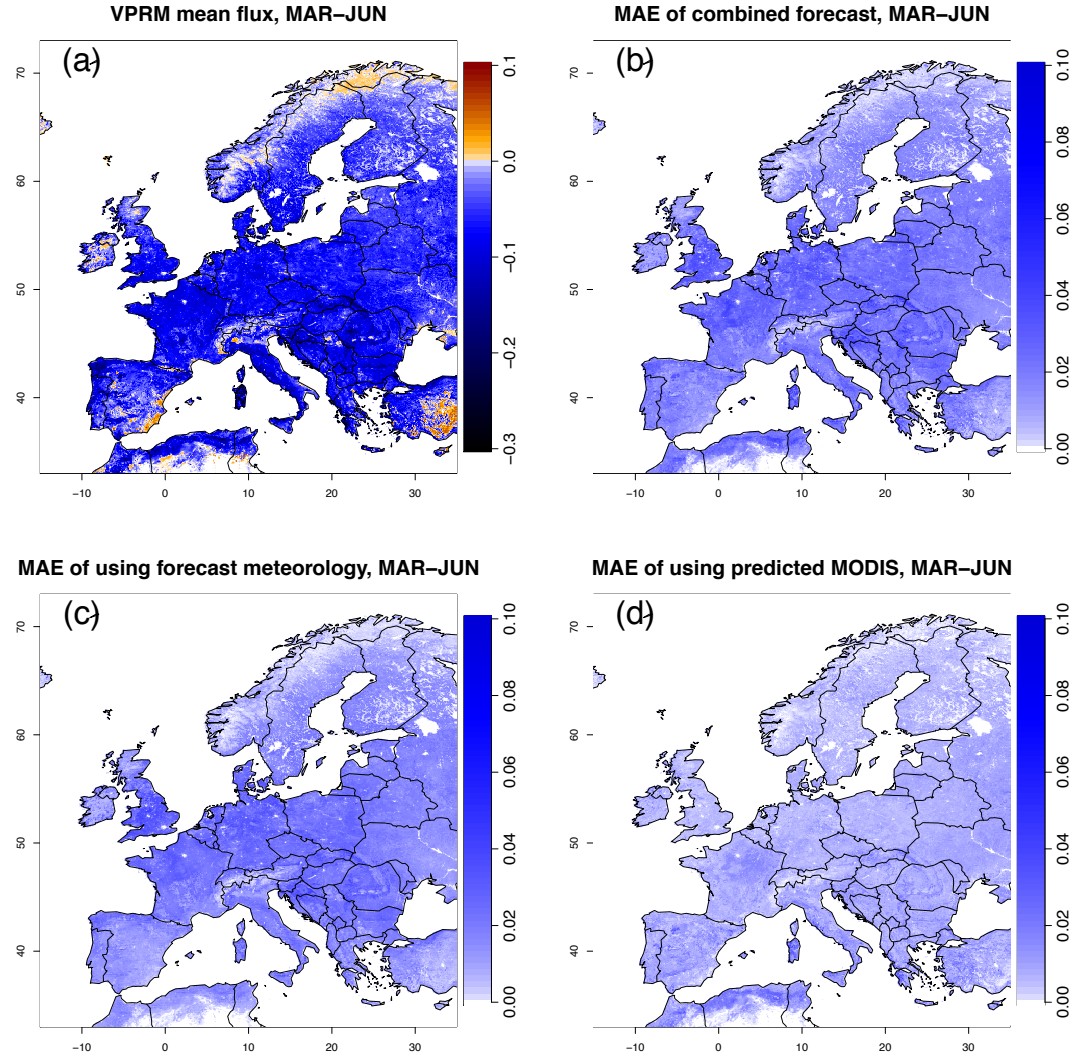


**Figure 10: (a) Mean VPRM NEE, during March to June 2014; (b) Spatial distribution of MAE for forecast error; (c) spatial distribution of MAE for meteorological error; (d) spatial distribution of MAE for MODIS error. The MAE of total forecast error in (b) has strong spatial relationship with the VPRM mean flux in (a), which indicates that the forecast error has a similar impact in all places. Panels (c) and (d) are**

**consistent with table 3, in that the forecast error is larger than the error from MODIS prediction.**

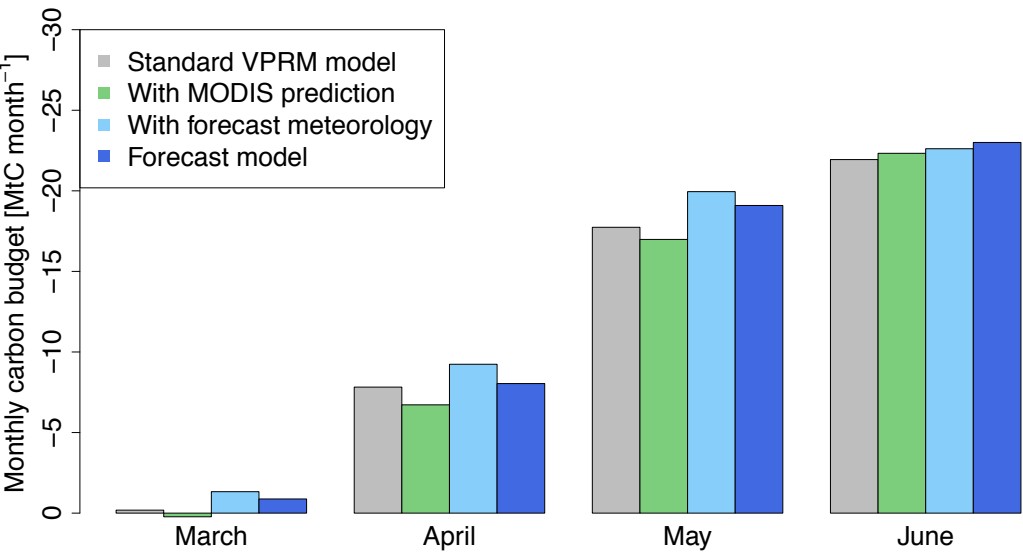

**Figure 11: Monthly carbon budget from March to June for original and forecast model for the European domain. The overall forecast flux budget is close to the original model, indicating the forecast flux model is appropriate for use in the GHG concentration forecasting system.**
