# Peer review of "Short-term forecasting of regional biospheric CO2 fluxes in Europe using a light-use-efficiency model (VPRM, MPI-BGC version 1.2)"

_Geoscientific Model Development, 2019_

## Short Comment (SC1) · 26 Jul 2019

Dear authors,

in my role as Executive editor of GMD, I would like to bring to your attention our Editorial version 1.2:

https://www.geosci-model-dev.net/12/2215/2019/

This highlights some requirements of papers published in GMD, which is also available on the GMD website in the 'Manuscript Types' section:

http://www.geoscientific-model-development.net/submission/manuscript_types.html

In particular, please note that for your paper, the following requirement has not been

met in the Discussions paper:

- "The main paper must give the model name and version number (or other unique identifier) in the title."

Please add a version number and the name / acronym of the light-use-efficiency model to the title,e.g:. Short-term forecasting of regional biospheric CO2 fluxes in Europe using a light-use-efficiency model (VPRM vX.Y).

Yours,

Astrid Kerkweg
* * *

---

## Short Comment (SC2) · 5 Aug 2019

Dear Astrid Kerkweg,

We thank you for reminding us the requirements of GMD, the version number is absolutely necessary for the traceability of the research.

Therefore we would revise the title as 'Short-term forecasting of regional biospheric CO2 fluxes in Europe using a light-use-efficiency model (VPRM, MPI-BGC version 1.2)', and also revise the corresponding main text to improve the clarity.

The reason behind the version name is that the VPRM model is originally developed in the paper Xiao et. al. 2004 and Mahadevan et.al. 2008. The principle and the algorithm of the model is easy to understand, so there is not a commonly used version in the

community but each group use their own version. Therefore we think this identification (VPRM, MPI-BGC version 1.2) would be sufficient to identify our own version.

Thanks again,

Jinxuan Chen
* * *

---

## Referee Comment (RC1) · Anonymous Referee #1 · 14 Oct 2019

This study uses a data driven model, VPRM, to forecast biosphere CO2 fluxes in a few days (5 days) in the future. With the forecasting shortwave radiation and temperature from a meteorological model, and the processed MODIS NRT EVI and LSWI, they forecasted the biosphere CO2 fluxes over Europe in 2014 and try to assess both the "model uncertainty" and "forecast uncertainty". They concluded that the forecasting error is less than the VPRM model error. The largest forecast error source comes from the meteorological data rather than MODIS inputs. The study is interesting and important for understanding the contribution of model uncertainty, especially the forecast uncertainty, for such data driven model. The research questions are clearly stated, and the figures are of excellent quality. However, some of the analysis seem to be not robust enough to support the conclusions. I have several major comments on this study: (1)

[Figure]

In this study, the "model error" of VPRM is estimated as the difference between the estimation from control simulation with perfect inputs and observed NEE flux in this study. However, this "model error" not only includes the error introduced by the VPRM model (input data, model parameters, and model structure), but also the error caused by the inconsistency of EC tower footprints (100-2000m, Baldocchi et al. 2001) and the spatial resolution of their simulation (10 x 10 km). Thus, the estimated result of "model error" in this study and their statement that "the error of the forecasting system is less than the VPRM model error" could be misleading. The authors at least need to show the landscape homogeneity in the 10 x 10 km surrounding of each EC tower sites used in this study, or to show the uncertainty caused by the GPP simulation at different spatial resolutions to the tower derived GPP. (2)When accounting for the error attribution from the meteorological variables, air temperature and downward shortwave radiation, they simply listed one site as an example and concluded that "it is the errors in shortwave radiation that mainly contribute to the meteorological data" (Figure 4). It would be more convincing if they can have a figure to show the distribution of GPPbias due to the bias of shortwave radiation (SWbias) and respiration (Rbias) accounting for all the sites. (3)There are already some studies to assess the uncertainty of the VPRM, for example, Lin et al. 2011, what are the similar or different conclusions between this study and Lin's? I suggest more discussion should be added in this paper. Minor comments: Line 30-31 Do you mean "carbon exchange between the surface and the atmosphere"? Line 207 What are those experimental simulations a to f? You need to refer to "Table 2" here and describe those simulations. Line 244-245 How do you calculate the "bias-GPP" and "bias-R"? Line 275 "an" should be "a" Page 17 The caption should appear above the table, and all the separators for "Latitude" and "Longitude" should be full stops rather than commas. Baldocchi, D., E. Falge, L. Gu, R. Olson, D. Hollinger, S. Running, P. Anthoni, C. Bernhofer, K. Davis, R. Evans, J. Fuentes, A. Goldstein, G. Katul, B. Law, X. Lee, Y. Malhi, T. Meyers, W. Munger, W. Oechel, K.T. Paw U, K. Pilegaard, H.P. Schmid, R. Valentini, S. Verma, T. Vesala, K. Wilson, and S. Wofsy, 2001: FLUXNET: A New Tool to Study the Temporal and Spatial Variability of Ecosystem-Scale Carbon

Dioxide, Water Vapor, and Energy Flux Densities. Bull. Amer. Meteor. Soc., 82, 2415–2434,https://doi.org/10.1175/1520-0477(2001)082<2415:FANTTS>2.3.CO;2 Lin, J.C., Pejam, M.R., Chan, E., Wofsy, S.C., Gottlieb, E.W., Margolis, H.A. and McCaughey, J.H., 2011. Attributing uncertainties in simulated biospheric carbon fluxes to different error sources. Global Biogeochemical Cycles, 25(2).

———————————————————

---

## Referee Comment (RC2) · Anonymous Referee #2 · 30 Apr 2020

The authors have analyzed the uncertainties on the vegetation photosynthesis and respiration model aimed for forecasting the 5 days biogenic CO2 uptake in conjunction with ECWMF weather forecast and MODIS satellite data. The comparison with eddy tower NEE flux at 31 sites over Europe is well organized and describes that meteorological data error has a largest contribution to producing error in NEE and no clear bias over land cover types. I really enjoyed much on reading this paper.

Minor Comments:

Page 4, Line 138-140: Generally, the respiration responds exponentially to temperature. But the authors use the liner function here. I guess that this would affect especially on the diurnal variation in respiration, though the error could be cancelled between daytime and nighttime. Also "vegetation respiration" should be "ecosystem respiration".

[Figure]

Page 4, Line 145: Write the long name for "alpha"

Page 5, Line 179: I like to know the difference between analysis and forecast. Are analysis for past, and forecast for future, though both are anyway estimated by same ECWMF model?

Page 6, Line 215: "the other simulations"

Page 6, Line 217: not "save", but "same"

Figure 4: Title of top panel y-axis should be "s"hortwave radiation.

Tables 3: MAE table for which item? "NEE" Clarify it.
* * *

---

## Author Comment (AC1) · 28 May 2020

We thank both reviewers for their positive comments and valuable suggestions on this work. We structure the point-by-point response as (1) the comments from referees in grey; (2) author's responses in black and (3) author's changes in manuscript, in which the line numbers refer to the revised manuscript.

**Response to Referee #1**

**Referee's comment:**
This study uses a data driven model, VPRM, to forecast biosphere CO2 fluxes in a few days (5 days) in the future. With the forecasting shortwave radiation and temperature from a meteorological model, and the processed MODIS NRT EVI and LSWI, they forecasted the biosphere CO2 fluxes over Europe in 2014 and try to assess both the "model uncertainty" and "forecast uncertainty". They concluded that the forecasting error is less than the VPRM model error. The largest forecast error source comes from the meteorological data rather than MODIS inputs. The study is interesting and important for understanding the contribution of model uncertainty, especially the forecast uncertainty, for such data driven model. The research questions are clearly stated, and the figures are of excellent quality. However, some of the analysis seem to be not robust enough to support the conclusions. I have several major comments on this study:

**Response:** Thank you for your positive feedbacks and valuable suggestions. With the following response, we hope that we have addressed all the concerns in the major and minor comments.

**Referee's comment:**
(1). In this study, the "model error" of VPRM is estimated as the difference between the estimation from control simulation with perfect inputs and observed NEE flux in this study. However, this "model error" not only includes the error introduced by the VPRM model (input data, model parameters, and model structure), but also the error caused by the inconsistency of EC tower footprints (100-2000m, Baldocchi et al. 2001) and the spatial resolution of their simulation (10 x 10 km). Thus, the estimated result of "model error" in this study and their statement that "the error of the forecasting system is less than the VPRM model error" could be misleading. The authors at least need to show the landscape homogeneity in the 10 x 10 km surrounding of each EC tower sites used in this study, or to show the uncertainty caused by the GPP simulation at different spatial resolutions to the tower derived GPP.

**Response:** Thank you for pointing this out. We agree that the statement might be misleading, due to the missing information in the manuscript. We would like to briefly clarify this issue here, and we revised the manuscript accordingly to improve its clarity.

In line 188-190 of the original manuscript, we mentioned that the evaluation and comparison was done at two spatial levels: at the flux observation site level and at the European domain level with 10 km x 10 km resolution. The evaluation at the observation site level uses a VPRM-point model in which the same formula

for GPP and R apply, yet, the model does not run with 10x10 km resolution grids. It uses site-measured meteorological variables and the site-labeled vegetation type as input to simulate NEE at the exact location of the EC tower.

Regarding the flux observation, we agree that the footprint would vary due to meteorological conditions, thus the measurement may represent NEE of different areas. Hollinger and Richardson (2005) attribute the random error in flux measurement to three reasons: The error associated with measurement system, the error associated with turbulent transport and the statistical error relating to footprint heterogeneity. They establish a method for flux measurement error estimation and analyze it on a half-hourly time scale. Chevallier et al. (2012)calculate the flux measurement uncertainty on a daily time scale based on hourly uncertainty estimation from Lasslop et al. (2008), and conclude that the daily uncertainty is small compared to the daily NEE magnitude. A similar approach is used in Broquet et al. (2013), where the uncertainty of daily flux measurement is ignored in observation-model comparison. Therefore in our study, all the comparisons are done at a daily timescale to minimize the flux measurement uncertainty.

Since the flux measurement uncertainty is small (according to the above discussion), we define the 'model error' as the mismatch between the flux measurement and the reference simulation. This 'model error' includes the error associated with misrepresentation of the vegetation processes, as well as the spatial representation error. The spatial representation error is also attributed as 'model error', because it is the model that is not capable enough to represent the flux over the (varied) footprint. The precise conclusion in the manuscript should be that 'the error of the forecasting system is less than the VPRM model error when comparing at site level'.

When discussing the other level of comparison over the European domain, we agree that more spatial representation error is introduced by the spatial averaging of each variable. It is an important error component and has to be mentioned in the manuscript. However at this level of aggregation we did not compare directly between the model and the observation.

The consideration behind our 'forecasting-control' and 'control-observation' comparison is as follows: we aim to find a criterion to evaluate the 'forecasting error', and the 'model error' is chosen here as this criterion. With the conclusion that 'the error of the forecasting system is less than the VPRM model error when comparing at site level', we indicate that the error added by the forecasting system is small compared to the inherent error in VPRM itself. Similar to the model error, the spatial representation error is also inherent to the error in the VPRM simulation. Quantifying this error would be possible, but is beyond the scientific scope of the current study.

**Changes in the manuscript:**
We added a more detailed discussion in section 2.2 (line 198-232) about the errors in the model. We modified lines 233-244 to better describe the how we compare the 'forecast error' against the 'model error'.

**Referee's comment:**

(2). When accounting for the error attribution from the meteorological variables, air temperature and downward shortwave radiation, they simply listed one site as an example and concluded that "it is the errors in shortwave radiation that mainly contribute to the meteorological data" (Figure 4). It would be more convincing if they can have a figure to show the distribution of GPPbias due to the bias of shortwave radiation (SWbias) and respiration (Rbias) accounting for all the sites.

**Response:** Thank you for the suggestion. To better isolate the error caused by the forecast error in shortwave radiation, we added two further experiments, b.1 and b.2. In b.1 only the shortwave radiation uses the 5-day forecast, while all other variables are from the control simulation. b.2 is similar to b.1 but testing for the affect of temperature. The following figure shows the bias distribution of the two experiments, the vertical spread of bias in (a) is slightly larger than (b). The overall normalized MAE of using forecast SW only is 0.053 while the normalized MAE of using forecast temperature only is 0.042. Therefore, we reside the conclusion as follows: 'Among the two variables from meteorological forecast, the error caused by the shortwave radiation is slightly larger than the error caused by temperature.'

[Figure]

**Changes in the manuscript:**
We added the above figure as figure 4 in the manuscript and modified the corresponding discussion (line 321-327).

**Referee's comment:**

(3). There are already some studies to assess the uncertainty of the VPRM, for example, Lin et al. 2011, what are the similar or different conclusions between this study and Lin's? I suggest more discussion should be added in this paper.

**Response:** We mentioned the work of Lin et al. (2011) in line 104-106 in the original manuscript, but we agree that a more detailed discussion should be added since Lin's work is very important for the methodology of error attribution and for understanding the uncertainty of the VPRM model.

In brief, Lin et al. (2011) established a general framework to attribute error to different uncertainty sources (driving data, model parameters, observations and model misrepresentation). In their work the model's sensitivity to each uncertainty sources is calculated. With an estimation of errors in each variable (input data, parameter etc.), one can then attribute the total error to those uncertainty sources by multiplying the error in the source with the model's sensitivity.

Back to this study, our target is to investigate the feasibility of using such a data-driven model to predict near-future carbon fluxes. Given the uncertainties in meteorological forecasts, the near-real-time MODIS product, and all the necessary extrapolations, it was not clear if such a model could still predict realistic carbon fluxes. Furthermore it was unclear what impact different extrapolation techniques might have for the short-term forecast of the MODIS indices in such an application. This study focused on these questions in particular, and was able to quantify the uncertainty arising from each of these factors when using VPRM for the prediction of near-future carbon fluxes.

**Changes in the manuscript:** We have added the above discussion in the introduction (line 108-121).

**Referee's comment:**
Minor comments: Line 30-31 Do you mean "carbon exchange between the surface and the atmosphere"?
**Response:** Has been corrected to 'atmosphere'.

**Referee's comment:**
Line 207 What are those experimental simulations a to f? You need to refer to "Table 2" here and describe those simulations.
**Response:** We have revised the sentence (line 258-259) to clarify this.

**Referee's comment:**
Line 244-245 How do you calculate the "bias-GPP" and "bias-R"?
**Response:** The VPRM has output of GPP and R for each experiment. Here bias-GPP is the difference between the GPP from simulation b to GPP from the control simulation, normalized by the range of annual NEE at each site; while *Bias-R* is the difference between the respiration from simulation b to respiration from the control simulation, also normalized by the range of annual NEE at each site. They use the same normalization scalar so that they are additive and comparable to $Bias_{NEE}$:

$$Bias_{NEE} = Bias_{-GPP} + Bias_R$$

$Bias_{-GPP}$ and $Bias_R$ represent the fractional bias of photosynthetic and non-photosynthetic part in NEE.

**Changes in manuscript:** We have add the definition of $Bias_{-GPP}$ and $Bias_R$ in section 2.2 (line 286-293) to improve the clarity.

**Referee's comment:**

**Response:** Thank you, we have corrected the mistake.

**Referee's comment:**
Page 17 The caption should appear above the table, and all the separators for "Latitude" and "Longitude" should be full stops rather than commas.
**Response:** We have corrected the tables.

**Response to Referee #2**

**Referee's comment:**
The authors have analyzed the uncertainties on the vegetation photosynthesis and respiration model aimed for forecasting the 5 days biogenic CO2 uptake in conjunction with ECWMF weather forecast and MODIS satellite data. The comparison with eddy tower NEE flux at 31 sites over Europe is well organized and describes that meteorological data error has a largest contribution to producing error in NEE and no clear bias over land cover types. I really enjoyed much on reading this paper.
**Response:** Thank you for your positive feedback.

**Referee's comment:**
Minor Comments: Page 4, Line 138-140: Generally, the respiration responds exponentially to temperature. But the authors use the liner function here. I guess that this would affect especially on the diurnal variation in respiration, though the error could be cancelled between daytime and nighttime.
**Response:** Yes, we agree that the temperature dependence of respiration is usually exponential, and a linear function would cause error in respiration estimation. Thus the capability of flux prediction for the VPRM model can potentially be improved. However in this study, we aim to test the current version of VPRM, a widely-used flux model in atmospheric CO2 simulation and inversion, for its capability to do flux forecasting. Therefore, we prefer and need to use the original version of the model from Mahadevan et al. (2008), so that other VPRM users can refer the results of this study.

**Referee's comment:**
Also "vegetation respiration" should be "ecosystem respiration".
**Response:** Thank you. We have replaced "vegetation respiration" with "ecosystem respiration".

**Referee's comment:**
Page 4, Line 145: Write the long name for "alpha"

**Response:** Done.

**Referee's comment:**

Page 5, Line 179: I like to know the difference between analysis and forecast. Are analysis for past, and forecast for future, though both are anyway estimated by same ECWMF model?

**Response:** The ECMWF uses their numerical weather prediction model IFS model for weather forecasting. The 'forecast' in this study refers to the operational forecast archive of ECMWF, which is the archive of the pure output of the IFS model at the corresponding time. On the other hand, due to the nonlinear characteristics of the atmosphere, the error in weather forecasting significantly increases as the model predicts longer into the future. Therefore the operational center always needs to optimize or constrain the model with meteorological observations from all over the world. Such optimization or so-called data assimilation can reduce the accumulated error in weather prediction, and constrains the model state closer to the real atmosphere conditions. The term 'analysis' in meteorology refers to this optimized model output.

**Changes in manuscript:** We have added a more detailed description for each error source in section 2.2 (line 222-232).

Page 6, Line 215: "the other simulations"

**Response:** Thank you, we have corrected the mistake.

Page 6, Line 217: not "save", but "same"

**Response:** Thank you, we have corrected the mistake.

Figure 4: Title of top panel y-axis should be "s"hortwave radiation.

**Response:** Thank you. As suggested by referee #1, we replace this figure to a new one that can better show the error contributions from shortwave radiation and temperature.

Tables 3: MAE table for which item? "NEE" Clarify it.

**Response:** Thank you, we have corrected the table.

**References**

Broquet, G., Chevallier, F., Bréon, F.-M., Kadygrov, N., Alemanno, M., Apadula, F., Hammer, S., Haszpra, L., Meinhardt, F., and Morguí, J.: Regional inversion of CO2 ecosystem fluxes from atmospheric measurements: reliability of the uncertainty estimates, 2013.

Chevallier, F., Wang, T., Ciais, P., Maignan, F., Bocquet, M., Altaf Arain, M., Cescatti, A., Chen, J., Dolman, A. J., and Law, B. E.: What eddy-covariance measurements tell us about prior land flux errors in CO2-flux inversion schemes, Global Biogeochem Cy, 26, 2012.

Hollinger, D., and Richardson, A.: Uncertainty in eddy covariance measurements and its application to physiological models, Tree physiology, 25, 873-885, 2005.

Lasslop, G., Reichstein, M., Kattge, J., and Papale, D.: Influences of observation errors in eddy flux data on inverse model parameter estimation, 2008.

Lin, J. C., Pejam, M. R., Chan, E., Wofsy, S. C., Gottlieb, E. W., Margolis, H. A., and McCaughey, J. H.: Attributing uncertainties in simulated biospheric carbon fluxes to different error sources, Global Biogeochem Cy, 25, Artn Gb2018 10.1029/2010gb003884, 2011.

Mahadevan, P., Wofsy, S. C., Matross, D. M., Xiao, X. M., Dunn, A. L., Lin, J. C., Gerbig, C., Munger, J. W., Chow, V. Y., and Gottlieb, E. W.: A satellite-based biosphere parameterization for net ecosystem $CO_2$ exchange: Vegetation Photosynthesis and Respiration Model (VPRM), Global Biogeochem Cy, 22, Artn Gb2005 10.1029/2006gb002735, 2008.